# Chemical Composition, Antibacterial Properties, and Anti-Enzymatic Effects of *Eucalyptus* Essential Oils Sourced from Tunisia

**DOI:** 10.3390/molecules28207211

**Published:** 2023-10-21

**Authors:** Sana Khedhri, Flavio Polito, Lucia Caputo, Vincenzo De Feo, Marwa Khamassi, Oumayama Kochti, Lamia Hamrouni, Yassine Mabrouk, Filomena Nazzaro, Florinda Fratianni, Ismail Amri

**Affiliations:** 1Laboratory of Management and Valorization of Forest Resources, National Institute of Researches on Rural Engineering, Water and Forests, Ariana 2080, Tunisia; sanakhedhrii@gmail.com (S.K.); khammassi_marwa@yahoo.fr (M.K.); hamrounilam@yahoo.fr (L.H.); 2Department of Pharmacy, University of Salerno, Via Giovanni Paolo II, 132, 84084 Fisciano, Italy; lcaputo@unisa.it (L.C.); defeo@unisa.it (V.D.F.); 3Institute of Food Science, National Research Council, Via Roma, 60, 83100 Avellino, Italyflorinda.fratianni@isa.cnr.it (F.F.); 4Laboratory of Biotechnology and Nuclear Technology, National Center of Nuclear Science and Technology, Sidi Thabet, Ariana 2020, Tunisia; kochtimayma@gmail.com (O.K.); mabrouk.yassine@cnstn.rnrt.tn (Y.M.); amri_amri@live.fr (I.A.)

**Keywords:** *Eucalyptus*, essential oil, biofilm, cholinesterase, α-amylase, α-glucosidase

## Abstract

This study was conducted to examine the chemical composition of the essential oils (EOs) from six Tunisian *Eucalyptus* species and to evaluate their anti-enzymatic and antibiofilm activities. The EOs were obtained through hydro-distillation of dried leaves and subsequently analyzed using GC/MS. The main class of compounds was constituted by oxygenated monoterpenes, particularly prominent in *E. brevifolia* (75.7%), *E. lehmannii* (72.8%), and *E. woollsiana* (67%). Anti-enzymatic activities against cholinesterases, α-amylase, and α-glucosidase were evaluated using spectrophotometric methods. Notably, the *E. brevifolia*,* E. extensa*,* E. leptophylla*,* E. patellaris*, and *E. woollsiana* EOs displayed potent acetylcholinesterase (AChE) inhibition (IC_50_: 0.25–0.60 mg/mL), with *E. lehmannii* exhibiting lower activity (IC_50_: 1.2 mg/mL). *E. leptophylla* and *E. brevifolia* showed remarkable α-amylase inhibition (IC_50_: 0.88 mg/mL), while *E. brevifolia* and *E. leptophylla* significantly hindered α-glucosidase (IC_50_ < 30 mg/mL), distinguishing them from other EOs with limited effects. Additionally, the EOs were assessed for their anti-biofilm properties of Gram-positive (*Staphylococcus aureus* and *Listeria monocytogenes*) and Gram-negative (*Acinetobacter baumannii*, *Pseudomonas aeruginosa* and *Escherichia coli*) bacterial strains. The *E. extensa* EO demonstrated the main antibiofilm effect against *E. coli* and *L. monocytogenes* with an inhibition > 80% at 10 mg/mL. These findings could represent a basis for possible further use of *Eucalyptus* EOs in the treatment of human microbial infections and/or as a coadjutant in preventing and treating Alzheimer’s disease and/or diabetes mellitus.

## 1. Introduction

The rise of antimicrobial resistance in recent years highlights a critical global health concern. This emergence of resistance underscores the urgency for innovative approaches in combating bacterial infections. In this context, the potential significance of natural products, particularly essential oils (EOs), becomes apparent in the quest to identify and develop novel antibacterial agents [1].

EOs have garnered attention due to their complex compositions, which often encompass a diverse array of bioactive compounds. Their multifaceted nature offers a unique advantage in circumventing bacterial resistance mechanisms, making them promising candidates for the development of effective antibacterial agents.

Furthermore, EOs have attracted attention beyond their direct antimicrobial properties. In recent times, research has unveiled their capacity to inhibit acetylcholinesterase (AChE) and/or butyrylcholinesterase (BChE), which are enzymes that play pivotal roles in neurodegenerative disorders such as Alzheimer’s disease [2,3].

This dual functionality of EOs, both as antimicrobial agents and potential contributors to neurodegenerative disease management, underscores their versatility and the value of exploring their various applications.

Moreover, the plant kingdom offers a rich reservoir of compounds with diverse therapeutic potentials. Among these potentials, the search for natural substances with hypoglycemic activities, notably α-glucosidase and α-amylase inhibitors, has gained interest. These enzymes are involved in the regulation of glucose levels, and their inhibition holds promise in managing conditions like diabetes [4,5]. The exploration of botanical sources for such bioactive compounds opens doors to developing new strategies for diabetes management and prevention.

As part of this context, the *Eucalyptus* genus, belonging to the Myrtaceae family, has attracted significant scientific interest with its extensive diversity of species and diverse biological activities, like antioxidant [6], anti-diabetic [7], phytotoxic [8] antimicrobial [9], and insecticidal [10] properties.

Moreover, many *Eucalyptus* species show antimicrobial activity: Elaissi and coworkers studied the antibacterial activity of 20 Eucalyptus species, among them, *E. odorata* EO showed the best inhibition zone diameter against *S. aureus* and *E. cinera* and *E. gunni* essential oils were the most active against *P. aeruginosa* [9]. Other Eucalyptus species like *E. gunnii* inhibited the biofilm formation of *S. aureus* [11]. Moreover, *E. globulus* was active against the biofilms of *E. coli* and *P. aeruginosa* [8] and showed a high potential to treat *Streptococcus pneumoniae* infections [12], while *E. bicostata* EO was reported to inhibit *A. baumannii*,* S. aureus*, and *L. monocytogenes* biofilms [13].

Regarding the anti-diabetic activity of EOs, plants from the *Eucalyptus* genus such as *E. globulus* have been used in the traditional treatment of diabetes [7]; moreover, *E. camaldulensis EO* inhibits both α-amylase and α-glucosidase in a non-competitive manner [14].

Although, recently, many EOs have been studied for their possible anti-cholinesterases activity, only two studies have been carried out in this direction for *Eucalyptus* species and regarding only *E. globulus* [15,16]. This study aimed to determine the chemical composition of the EOs of six Tunisian *Eucalyptus* species and to evaluate their antibacterial and anti-enzymatic activity. The six selected species are as follows: *Eucalyptus brevifolia* F. Muell., *E. extensa* L.A.S. Johnson and K.D. Hill, *E. lehmannii* (Schauer) Benth., *E. leptophylla* F. Muell. ex Miq., *E. patellaris* F. Muell., and *E. woollsiana* R.T. Baker. The antibiofilm activity was evaluated against Gram-positive (*S. aureus* and *L. monocytogenes*) and Gram-negative (*E. coli*, *P. aeruginosa*, and *A. baumannii*) pathogenic strains. The anti-enzymatic activity was evaluated against cholinesterases, α-amylase, and α-glucosidase.

## 2. Results

### 2.1. Yields and Chemical Composition

The hydro-distillation process of *Eucalyptus* leaves resulted in the extraction of yellow oils with yields of 2.42%, 2.03%, 1.91%, 1.66%, 1.43%, and 0.25% for *E. brevifolia*, *E. leptophylla*, *E. lehmannii*, *E. woollsiana*,* E. extensa*, and *E. patellaris*, respectively.

The analysis of the EOs allowed us to identify a total of 139 components: 40 in the *E. brevifolia* EO (97.2% of the total), 46 in the *E. extensa* EO (97.4%), 7 in the *E. lehmanii* EO (95.3%), 42 in the *E. lephophylla* EO (97.5%), 65 in the *E. patellaris* EO (94.1%), and 53 in the *E. woollsiana* EO (97.0%). In almost all EOs, oxygenated monoterpenes were the most representative class: 75.7, 58.8, 72.8, 39.2, and 67.0%, respectively, for the *E. brevifolia*,* E. extensa*,* E. lehmanii*,* E. leptophylla*, and *E. woollsiana* EOs. Instead, oxygenated sequiterpenes predominated in the EO from *E. patellaris* (46.2%) (Table 1).

In the *E. brevifolia* EO, in addition to oxygenated monoterpenes, monoterpenes hydrocarbons (19.6%), oxygenated sesquiterpenes (0.7%), sesquiterpenes hydrocarbons (0.2%), and compounds from different classes (1.0%) were present. The main component was eucalyptol (57.0%) followed by α-pinene (16.6%). Other compounds present in amounts of 1% or greater were *trans*-pinocarveol (5.8%), α-terpineol (4.1%), borneol (1.9%), pinocarvone (1.7%), *exo*-fenchol (1.2%), γ-terpinene (1.1%), and camphene (1.0%).

In the EO of *E. extensa*, in addition to oxygenated monoterpenes, oxygenated sesquiterpenes (16.5%), hydrocarbon monoterpenes (14.1%), and hydrocarbon sesquiterpenes (8.0%) were present. The main components were eucalyptol (50.0%) and α-pinene (12.0%). Other compounds present in amounts of 1% or greater were globulol (3.9%), β-eudesmol (3.2%), aromadendrene (2.9%), γ-eudesmol (2.9%), α-eudesmol (2.8%), α-terpinyl acetate (2.4%), spathulenol (2.3%), *trans*-pinocarveol (2.1%), α-terpineol (1.6%), and α-phellandrene (1.0%). 

Only monoterpene hydrocarbons (22.5%) have been detected in the EO of *E. lehmanii*; in addition to the oxygenated monoterpenes, eucalyptol (70.5%) and α-pinene (21.4%) are the main components (1.3%). 

In the *E. leptophylla* EO, in addition to oxygenated monoterpenes, hydrocarbon sesquiterpenes (15.9%), oxygenated sesquiterpenes (8.9%), hydrocarbon monoterpenes (0.8%), and other classes different from these (32.7%) were present. The main component was eucalyptol but in lower quantities than the other EOs (32.2%), followed by 2-methylbutanoic anhydride (31.0%). Other components present in amounts of 1% or greater were germacrene B (10.2%), *trans*-pinocarveol (4.6%), aromadendrene (4.4%), 10-*epi*-cubebol (3.4%), globulol (1.2%), and rosifoliol (1.0%). 

In the *E. patellaris* EO, in addition to oxygenated monoterpenes and sesquiterpenes, monoterpene hydrocarbons (7.4%), sesquiterpene hydrocarbons (5.5%), and other compounds from different classes (0.3%) were present. The main component, unlike what was detected in the other OEs studied in this work, was not eucalyptol (26.3%) but spathulenol (28.3%). Other components at or above 1% were β-eudesmol (3.9%), cyperotundone (3.0%), terpinen-4-ol (2.1%), cedr-8(15)-en-9-α-ol (2.0%), eudesm-7(11)-en-4-ol (1.9%), 7-*epi*-α-eudesmol (1.8%), α-terpineol (1.6%), α-phellandrene and α-guaiene (1.5%), α-pinene (1.3%), and γ-terpinene (1.0%).

In the *E. woollsiana* EO, in addition to oxygenated monoterpenes, sesquiterpene hydrocarbons (12.2%), monoterpene hydrocarbons (9.0%), oxygenated sesquiterpenes (7.9%), and compounds from different classes (0.9%) were present. The main component was eucalyptol (53.6%). Other components present in amounts of 1% or greater were α-pinene (7.5%), aromadendrene (4.7%), *trans*-pinocarveol (4.6%), 10-*epi*-cubebol (4.6%), β-eudesmol (1.6%), pinocarvone (1.4%), α-amorphene (1.3%), γ-patchoulene (1.3%), α-cadinene (1.3%), and *allo*-aromadendrene (1.1%). 

### 2.2. Antibiofilm Activity

The investigated EOs demonstrated a robust inhibitory activity on biofilm, with MIC values ranging from 16 mg/mL to values exceeding 50 mg/mL, as delineated in Figure 1.

To evaluate the association between the EOs and their respective antibacterial properties, a hierarchical cluster analysis (HCA) was conducted using the mean MIC values. The antimicrobial effects of the EOs displayed significant variations among both *Eucalyptus* species and bacterial strains (*p* < 0.05). 

The HCA analysis resulted in the classification of two distinct EO groups, designated as Group A and Group B, based on their antibacterial efficacy, with a dissimilarity exceeding or equal to 12 (Figure 2).

Group A was primarily comprised of *E. extensa*, which stood out as a notable exception. This particular species exhibited strong antibacterial activity against all tested strains, with an MIC of 16 mg/mL, except for *A. baumanii* (MIC > 30 mg/mL).

Group B was further divided into three subgroups, labeled B1, B2, and B3, which collectively displayed moderate and stable antibacterial activity across all species.

In Subgroup B1, represented by *E. lehmannii*, the MIC against *L. monocytogenes* was 38 ± 4 mg/mL, while it showed moderate activity against *P. aeruginosa* (28 ± 2 mg/mL), *S. aureus* (30 ± 3 mg/mL), and 35 mg/mL for Gram-negative bacteria.

Subgroup B2, characterized by the EOs of *E. patillaris* and *E. leptophylla*, exhibited a moderate inhibitory effect against all tested bacterial strains. The *E. patellaris* EO demonstrated a more pronounced impact on Gram-positive bacteria, with an MIC of 28 mg/mL, compared with its effects on Gram-negative strains such as *E. coli* (MIC = 30 ± 2 mg/mL) and *A. baumanii* (MIC = 35 ± 2 mg/mL). Conversely, the *E. leptophylla* EO displayed an MIC of 30 ± 2 mg/mL against *E. coli*, 30 ± 3 mg/mL against *L. monocytogenes*, and 28 mg/mL for the remaining bacterial strains.

Within Subgroup B3, characterized by the *E. woolsiana* and *E. brevifolia* EOs, limited antibacterial activity was observed against the tested strains. Both EOs exhibited the highest MIC values (>50) when tested against two Gram-negative bacteria (*E. woolsiana* vs. *E. coli* and *E. brevifolia* vs. *A. baumannii*). The lowest recorded MIC values were 32 ± 4 for *S. aureus* with *E. woolsiana* and 28 ± 2 for *E. coli* with *E. brevifolia*.

It is important to note that the tested EOs demonstrated lower activity when compared with tetracycline, as indicated in Figure 1. In terms of MIC values, the EOs rich in oxygenated sesquiterpenes like globulol, spathulenol, and β-eudesmol, exemplified by *E. extensa* (16.5%) and *E. patellaris* (46.2%), exhibited the lowest MIC values. 

The MIC values offered valuable insights into the potential of the EOs to influence bacterial biofilms. Additionally, an assessment was conducted to ascertain the capacity of the EOs to impact established biofilms and modify the metabolic processes of bacterial cells. The results are depicted in Table 2 and Table 3.

Each EO exhibited distinct inhibitory effects on biofilm, with efficacy varying based on bacterial species and concentrations. Apart from *A. baumannii*, mature biofilms displayed sensitivity to the impact of EOs, thus highlighting their receptiveness. Notably, *A. baumannii* demonstrated resistance when encountering *E. brevifolia* and *E. extensa*. However, noteworthy inhibition of cellular metabolism was observed at the highest concentration, resulting in percentages of inhibition of 28.83 ± 1.87 and 65.40 ± 4.57%, respectively. This underscores the distinct effectiveness of these oils in affecting microbial activity.

The efficacy of the EO from *E. leptophylla* was evident in reducing biofilm formation across all tested bacterial strains. Inhibition values ranged from 60.47 ± 4.54% (for *L. monocytogenes*) to 93.17± 1.02% (for *P. aeruginosa*) at 20 mg/mL. Furthermore, this impact extended to hindering the cellular metabolism of these pathogens, with inhibition values varying from 60.47 ± 3.08 (for *E. coli*) to 93.17 ± 1.13% (for *L. monocytogenes*) at the same concentration of 20 mg/mL.

Interestingly, the inhibition caused by the EOs did not consistently correspond to a comparable influence on the metabolic activity of bacterial cells within the biofilm. For instance, the *E. woollsiana* EO exhibited a notably higher effect on *A. baumannii* cellular metabolism, reaching 93.27 ± 1.01%, compared with its impact on mature biofilm (37.59 ± 2.47%) at the identical concentration (20 mg/mL).

Similarly, the *E. extensa* EO demonstrated a substantial (73.73 ± 3.37%) inhibition of *S. aureus* biofilm. However, the inhibition of cellular metabolism resulted in only 11.30 ± 1.07% at 20 mg/mL.

### 2.3. Anti-Enzymatic Activity

Table 4 shows the anti-enzymatic activity of the EOs. The *E. brevifolia*, *E. extensa*, *E. leptophylla*, *E. patellaris*, and *E. woollsiana* EOs showed similar activity against AChE with IC_50_ values ranging from 0.25 to 0.60 mg/mL; the least active EO was *E. lehmanii*.

Regarding the EO activity against BChE, the *E. lehmanii* and *E. patellaris* EOs showed similar activity with IC_50_ values of 3.48 and 3.50 mg/mL, respectively. The activity of the other EOs was in the following order: *E. woollsiana* > *E. leptophylla* > *E. extensa* > *E. brevifolia*.

The *E. leptophylla*, *E. brevifolia,* and *E. patellaris* EOs were the most active against α-amylase followed by the *E. woollsiana* and *E. lehmanii* EOs. Moreover, the *E. brevifolia* and *E. leptophylla* EOs were active against α-glucosidase but the other EOs showed no activity against this enzyme with IC_50_ > 30 mg/mL.

## 3. Discussion

This investigation has elucidated discernible variations in both the yield and chemical composition of EOs, a phenomenon influenced by a multitude of factors encompassing both external and internal variables. Among the external factors, environmental conditions such as precipitation, temperature, light exposure, soil composition, and altitude exert significant influence over the growth of aromatic plants and the subsequent biosynthesis of their EOs. Furthermore, the environment can impart modifications to the plant’s genetic makeup, thereby introducing variations in its genotype [17].

These pronounced disparities in EO yields are conspicuously evident across all *Eucalyptus* species. To provide a specific illustration, *E. brevifolia* emerges as the frontrunner in EO yield, boasting an impressive 2.42%, while *E. patellaris* lags behind with a considerably lower yield of only 0.25%.

Upon undertaking a comparative analysis with previous research, it becomes increasingly apparent that a substantial paucity of available data pertains to the EO yields of *E. extensa*, *E. leptophylla*, and *E. woollsiana*. In contrast, the EO yields of *E. patellaris* in our study (0.25%) align closely with the findings documented by Elaissi et al. [18], which exhibit a range from 0.1% (January) to 0.5% (June).

Meanwhile, our research reveals a higher yield for *E. brevifolia* at 2.42%, in contradistinction to the yield reported by Ben Hassine et al. [18], which amounted to 1.5%. A parallel observation in the case of *E. lehmannii*, as our study attests to a yield of 1.91%, surpassed the yield of 1.25% as reported by Ben Slimane et al. [19] and closely approached the value recorded (2.21 ± 0.24) by Jemâa et al. [20].

Recently, in a study conducted by Ben Hassine and colleagues [21], the petroleum ether extract of *E. brevifolia* was analyzed using GC-MS. Their research revealed a composition primarily characterized by high levels of eucalyptol and spathulenol, followed by globulol and epiglobulol. However, it is important to note that specific percentages were not provided in their report. These findings align with the composition of our sample only for the presence of eucalyptol as the dominant component. In fact, while globulol, spathulenol, and epiglobulol were also detected in the *E. brevifolia* EO studied in our work, they were present in very small quantities of 0.4%, 0.1%, and 0.1%, respectively.

In contrast, another study by Ben Hassine and collaborators [22] examined the *E. brevifolia* EO characterized by a markedly different composition. This EO was notably rich in germacrene D-4-ol (23.5%), eucalyptol (14.0%), 2-butoxyethyl acetate (9.4%), *trans*-2,8-menthdienol (7.6%), α-santalene (8.0%), tricyclene (6.5%), ipsdienone (5.0%), *trans*-nerolidol (2.4%), *trans*-isolimonene (2.0%), cedrol (1.8%), elemol (1.6%), 3,5-acoradien-11-ol (1.6%), hinesol (1.5%), *o*-cymene (1.4%), α-guaiene (1.3%), longifolene (1.2%), and β-chamigrene (1.0%). This composition differed significantly from the one observed in our study. Only eucalyptol and longifolene were present in the EO examined in our study, although in different proportions. Eucalyptol was the primary component in our sample (57.0%), but it was present in a yield of only 14.0% in the EOs studied by Ben Hassine and coworkers. Conversely, longifolene was present in higher amounts (1.2%) than in our *E. brevifolia* EO (0.1%). It is worth noting that the other major compounds identified in Ben Hassine’s research differed both in terms of their chemical composition and quantity when compared with our study.

In the EO of *E. extensa,* oxygenated monoterpenes are the main compounds (58.8%) with eucalyptol (50.0%) as the principal compound; the other constituents are oxygenated sesquiterpenes (16.5%), hydrocarbon monoterpenes (14.1%), and hydrocarbon sesquiterpenes (8.0%). No previous studies have reported the chemical composition of *E. extensa* EO.

Conversely, the literature concerning *E. lehmannii* has been enriched with noteworthy contributions, particularly in the context of Tunisian species. A previous study delved into the composition of *E. lehmannii* EOs during various seasons, revealing camphene (21.1%), 1,8 cineole (18.4%), α-terpineol (15.1%), α-pinene (7.2%), and *trans*-pinocarveol (5.1%) as the predominant components [20]. However, the *E. lehmannii* EO examined in the current study displayed eucalyptol as the predominant component (70.5%), followed by α-pinene (21.4%), while camphene represents a mere 0.9% and α-terpineol only 0.1% of the composition. Our results corroborate with previous studies that reported eucalyptol and α-pinene as major constituents [19,23], even if the EOs analyzed in these studies presented a more intricate composition compared with the EO under examination. In contrast, in the EO of *E. lehmannii* reported by Yangui et al. [24], the main constituent, eucalyptol, was present in notably comparable proportions, with a content of 67.2% in comparison with 70.5% of the present study. However, the remaining components diverge significantly between the two studies. Yangui’s exploration revealed a more extensive array of chemical constituents within the *E. lehmannii* EO, with α-gurjunene (6.5%) and β-panasinsene (4.2%) being notable among them, which were absent in the EO examined in the current study. Furthermore, α-pinene, the second most abundant component in the EO analyzed in this study (21.4%) also exists in Yangui’s EO but at a lesser relative abundance, constituting merely 2.1% of the composition.

Only one previous study reported the composition of the *E. leptophylla* EO [25]. This analysis revealed a complex composition featuring abundant eucalyptol (66.4%), α-pinene (5.8%), *trans*-pinocarveol (4.9%), aromadendrene (4.7%), globulol (2.3%), β-eudesmol (1.9%), pinocarvone (1.5%), *allo*-aromadendrene (1.2%), *trans*-*p*-mentha-1(7),8-dien-2-ol, and *cis*-p-mentha-1(7),8-dien-2-ol (1.0%). Striking similarities in major components emerge between this EO and our current study, although quantities frequently differ. Eucalyptol constitutes the main component in both, albeit in different percentages (66.4% vs. 32.2%), while α-pinene, *trans*-pinocarveol, and aromadendrene were detected in comparable percentages.

Insight into the composition of *E. patellaris* EO is limited. Bignell and Dunlop [26] reported aromadendrene (32.8%) as the major component followed by eucalyptol (22.3%), verbenone (10.4%), *allo*-aromadendrene (6.8%), globulol (3.1%), β-pinene (2.1%), α-terpineol (1.9%), α-pinene, spathulenol (1.2%), and β-selinene (1.0%). Even if some compounds align with our study, such as eucalyptol and α-pinene, substantial quantitative discrepancies exist. In a recent study by Elaissi and colleagues [18], the compositions of two distinct *E. patellaris* EOs derived from plants harvested during different periods were reported. The EO extracted from the July 2004 harvest predominantly consists of eucalyptol (11.1%), viridiflorol (7.3%), *p*-cymen-8-ol (5.6%), α-pinene (4.5%), *cis*-*p*-mentha-1,8-dien-6-ol (3.6%), pinocarvone (3.2%), β-gurjunene (2.7%), geraniol (2.1%), calachorene (1.9%), β-elemene (1.7%), palustrol (1.6%), *cis*-*p*-mentha-(7)-8-dien-2-ol (1.3%), linalool (1.1%), α-eudesmol (1.1%), and camphor (1.0%). In contrast, the EO extracted from the January 2005 harvest primarily contains eucalyptol (32.5%), viridiflorol (11.2%), limonene (8.9%), guaiol (7.0%), α-terpinene (6.1%), α-phellandrene (3.0%), α-pinene (2.8%), isobutyl isovalerate (2.1%), β-gurjunene (1.9%), α-eudesmol (1.6%), pinocarvone (1.5%), γ-terpinene (1.4%), and palustrol (1.0%). It is worth noting that these compositions exhibit only minimal similarities with the EO studied in this research. Of the two, the composition of the January EO bears the closest resemblance to the EO studied in this work. Both EOs share some major components (present in quantities exceeding 1%), including eucalyptol, which is the main component in the study by Elaissi (32.5%) and the second most prominent in this work (26.3%), α-pinene with varying amounts (2.8% in Elaissi’s study and 1.3% in this work), α-phellandrene (3.0% and 1.5%), and γ-terpinene in similar quantities (1.2% and 1.0%). Notably, the EO obtained from the July 2004 harvest has only two major components in common among those exceeding 1%, namely, eucalyptol (11.1% vs. 26.3%) and α-pinene (4.5% vs. 1.3%), with the former being the primary component in Elaissi’s EO but not in the EO researched in this study.

As for the chemical composition of the EOs of *E. extensa* and *E. woollsiana*, pertinent data remain conspicuously absent in the existing literature.

In summary, the variations in the percentage compositions imply a potential link between chemotypes (variations in chemical profiles) and genotypes (genetic characteristics) [27,28]. Indeed, these differences in composition are influenced by a combination of external environmental factors and internal genetic and physiological traits.

Undeniably, biofilms of pathogenic bacteria present a significant contemporary challenge across various sectors due to their inherent resistance to antibiotics, resulting in the persistence of chronic infections [29]. Consequently, the exploration of alternative treatments for biofilms has garnered considerable interest. *Eucalyptus*, known for its antibacterial properties, has recently been under scrutiny for its potential antibiofilm effects [30].

The potential of the tested EOs to combat biofilm formation appears to be relatively constrained. However, it is noteworthy to mention that Limam et al. [31] have previously documented moderate yet discernible antimicrobial properties associated with *E. lehmannii,* illustrating its effectiveness against pathogenic bacteria such as *Pseudomonas aeruginosa*, *Escherichia coli*, and *Staphylococcus aureus*, findings that align with our observations. Furthermore, the EO derived from *E. brevifolia* exhibited competence in combating Gram-negative bacterial strains [22]. It is worth highlighting that several *Eucalyptus* EOs have demonstrated promising attributes in inhibiting the formation of bacterial biofilms.

Mechanistic insights into the antibacterial efficacy of EOs encompass interactions with bacterial DNA, membrane permeability modulation, and alterations in membrane fluidity due to their hydrophobic nature. These actions can disrupt cellular integrity, releasing intracellular constituents and rendering them susceptible to biofilm-inhibiting agents [32]. Moreover, terpenes, particularly, can induce bacterial cell leakage, leading to the efflux of critical molecules and ions, ultimately resulting in cell death [33].

Considering the composition of the six EOs, it becomes plausible to establish a connection between their efficacies in restricting the virulence of pathogenic strains. This could be attributed to the interplay of various compounds acting concurrently on multiple targets through diverse mechanisms, either in synergy or antagonism [34].

In our study, eucalyptol represents a trait component for the tested *Eucalyptus* EOs. This molecule is a compound renowned for its antimicrobial activity against a broad spectrum of both Gram-positive and Gram-negative pathogens, including those featured in our experimental studies. According to LaSarre and Federle [35], the impact of eucalyptol on quorum-sensing mechanisms is notable, but it does not impede the vital functions of *A. baumannii*. In the case of *E. coli*, this compound could have initiated a robust condensation process within the nuclear chromatin of its bacterial nucleosome, while concerning *S. aureus*, the presence of eucalyptol might have triggered unintended apoptosis [36,37]. It is probable that differences between EO antimicrobial activities could be due to the different ratios of oxygenated monoterpenes and sesquiterpenes as reported by Martins and coworkers [38].

In light of these results among the tested EOs, the *E. extensa* EO could be considered for the development of a possible antimicrobial agent. In fact, *E. extensa* presented the lowest MIC value and was capable of inhibiting biofilm at lower concentrations with respect to the other tested essential oils, in particular, it was active against the mature biofilms of *E. coli* and *L. monocytogenes*.

Previous studies have explored the anti-AChE, anti-BChE, anti-α-amylase, and anti-α-glucosidase potential of other *Eucalyptus* EOs [7,16,39,40,41,42], but no previous studies on the species examined here are present in the available literature. The presence of 1,8-cineole in our samples likely contributed to EO activity against acetylcholinesterase; in fact, in a previous work, an IC_50_ of 13.5 µg/mL was reported for 1,8 cineole [42]. Moreover, 1,8 cineole displayed more robust inhibition against α-amylase, reducing enzyme activity by 43% at 75 µg/mL in comparison with α-glucosidase (IC_50_ 1 mg/mL) [16]. These findings might elucidate the disparity in anti-α-glucosidase activity relative to α-amylase activity in the studied EOs.

Significant AChE inhibitory activity has been reported for the EOs of other species like *E. camaldulensis* Dehnh., *E. intertexta* R.T. Baker, and *E. diversifolia* Bonpl. [24,25]. Moreover, *E. globulus* Labill. EO has demonstrated AChE and BChE inhibition [16]. The EOs from *E. globulus*, *E. citriodora* Hook., and *E. camaldulensis* displayed potent antidiabetic activity by curtailing postprandial hyperglycemia in type 2 diabetic rat models and inhibiting α-amylase and α-glucosidase activity [7,14].

Among the studied EOs, *E. lehmanii* showed the best inhibitory activities against cholinesterases. Furthermore, *E. brevifolia* and *E. leptophylla* could be taken into consideration as possible coadjutants in the treatment of diabetes in further studies.

## 4. Materials and Methods

### 4.1. Plant Materials

The plant material was collected during the spring season from arboretums affiliated with the National Institute of Research on Rural Engineering, Water, and Forests (Table 5). The species investigated included *Eucalyptus brevifolia* F. Muell., *E. extensa* L.A.S. Johnson and K.D. Hill., *E. lehmannii* (Schauer) Benth., *E. leptophylla* F. Muell. ex Miq., *E. patellaris* F. Muell., and E. woolsiana F. Muell. ex R.T. Baker.

To ensure representative samples, at least five different trees were sampled for each species, and their respective samples were combined and homogenized. The resulting homogenous samples were then placed in a greenhouse and dried in the shade for a period of 3–5 days, or until a constant weight was achieved.

The samples were authenticated by the herbarium division of the institute.

The extraction yields were determined by applying this formula:$$Yields=\frac{MEO\;*\;100}{DM}$$
where DM represents dry material and MEO stands for the mass of EO.

### 4.2. Isolation and Analysis of the Essential Oils

The EOs were obtained via hydro-distillation of dried leaf samples. The distillation process was carried out using a Clevenger apparatus and the EOs were collected and dried using anhydrous sodium sulfate, and then stored in sealed glass brown vials in a refrigerator at 4 °C until further studies.

A Perkin-Elmer Sigma-115 gas chromatograph (Perkin Elmer, Waltham, MA, USA) equipped with a flame ionization detector (FID) and a data handling processor was used for analytical gas chromatography (GC). The separation was achieved using an HP-5 MS fused silica capillary column (30 m, 0.25 mm i.d., 0.25 μm film thickness, Agilent, Roma, Italy). A column temperature of 40 °C was used, with a 5 min initial hold, followed by temperatures of 270 °C at 2 °C/min and 270 °C (20 min). The injection mode was splitless (1 μL of a 1:1000 *n*-hexane solution) with injector and detector temperatures of 250 °C and 290 °C, respectively. The analysis was also run using a fused silica HP Innowax polyethylene glycol capillary column (50 m, 0.20 mmi.d., 0.25 0.25 μm film thickness, Agilent, Roma, Italy). In both cases, helium was used as the carrier gas (1.0 mL/min).

The GC/MS analyses were performed on an Agilent 6850 Ser. II apparatus (Agilent, Roma, Italy) fitted with a fused silica DB-5 capillary column (30 m, 0.25 mm i.d., 0.33 μm film thickness, Agilent, Roma, Italy) and coupled to an Agilent Mass Selective Detector MSD 5973 with an ionization energy voltage of 70 eV and an electron multiplier voltage energy of 2000 V. Mass spectra (MS) were scanned in the range 40–500 amu with a scan time of 5 scans/s.

The majority of constituents were identified using GC by comparing their Kovats retention indices (Ri), which were determined relative to the retention times (tR) of *n*-alkanes (C10-C35), to either those in the literature [43,44] and mass spectra on both columns or those of authentic compounds available in our laboratories via the NIST 02 and Wiley 275 libraries. Peak area normalization was used to obtain the component relative concentrations. There were no response factors calculated.

### 4.3. Antimicrobial Activity

#### 4.3.1. Microorganisms and Culture Conditions

The following bacterial strains were utilized: *Acinetobacter baumannii* ATCC 19606, *Pseudomonas aeruginosa* DSM50,071, and *Escherichia coli* DSM 8579 (Gram-negative); *Staphylococcus aureus* subsp. aureus Rosebach ATCC 25923 and *Listeria monocytogenes* ATCC 7644 (Gram-positive). Prior to analysis, the bacteria were cultivated in Luria broth at 37 °C for 18 h. *A. baumannii* was cultured under the same conditions at 35 °C.

#### 4.3.2. Minimal Inhibitory Concentration (MIC)

To ensure sterility, the EOs and the DMSO underwent ultrafiltration before their use in the study. The MIC (Minimum Inhibitory Concentration) of the EOs was determined using a modified version of the resazurin method developed by Sarker and Nahar [45]. A resazurin solution was prepared by dissolving 270 mg of resazurin in 40 mL of sterilized deionized water. In 96-well microtiter plates, the first row received 100 μL of samples in DMSO (1:10 *v*/*v*), while all other wells received 50 μL of Luria–Bertani broth or normal sterile solution. Serial dilutions of the EOs were performed in descending concentrations. To each well, 10 μL of the resazurin indicator solution was added. Furthermore, 30 μL of 3.3 ×sensitized broth and 10 μL of bacterial suspension (5 × 10^6^ cfu/mL) were added to each well. The plates were sealed with parafilm to prevent dehydration. A column of the plate contained the broad-spectrum antibiotic tetracycline, which was previously suspended in DMSO and served as a positive control [14]. A negative control consisted of Luria–Bertani broth containing resazurin and bacteria without any samples. The plates were incubated at 37 °C (35 °C for *A. baumannii*) for 24 h. Visual observation was used to assess any color changes. If the solution changed from dark purple to pink or colorless, it was recorded as a positive result. The MIC value was determined as the lowest concentration that could prevent the color change from dark purple to pink.

#### 4.3.3. Biofilm Inhibitory Activity

To evaluate the inhibitory activity on mature biofilm, flat-bottomed 96-well microtiter plates were employed [46]. Bacterial cultures were adjusted to a 0.5 McFarland standard with fresh culture broth. Each well received 10 μL of the bacterial cultures and then they were incubated for 24 h at 37 °C (35 °C for *A. baumannii*). After removing the planktonic cells, in each well 10 or 20 μL/mL of the EOs were added. The final volume in each well was adjusted to 250 μL with varying amounts of Luria–Bertani broth. The plates were covered with parafilm tape to prevent evaporation and incubated at 37 °C (35 °C for *A. baumannii*) for another 24 h. After removing the planktonic cells, sessile cells were washed twice with sterile PBS. Subsequently, the plates were left under a laminar flow hood for 10 min to fix the sessile cells and then removed after 15 min. The plates were allowed to dry and the sessile cells were stained with 200 μL of a 2% *w*/*v* crystal violet solution per well for 20 min. Then, the staining solution was discarded and the plates were gently washed with sterile PBS. The bound dye was released by adding 200 μL of 20% *w/v* glacial acetic acid. The absorbance was measured at λ = 540 nm using a spectrophotometer (Cary Varian, Palo Alto, CA, USA). The biofilm inhibitory activity was calculated as a percentage relative to the control (cells grown without the samples were considered to have 0% inhibition). Triplicate tests were performed and average results were calculated for reproducibility.

#### 4.3.4. Effects of EOs on Cell Metabolic Activity within Biofilm

To evaluate the effect of the EOs on the metabolic activity of bacterial cells within the biofilm, the 3-(4,5-dimethylthiazol-2-yl)-2,5-diphenyltetrazolium bromide (MTT) colorimetric method was employed [46]. Two concentrations of the EOs (10 and 20 μL/mL) were added after 24 h of bacterial incubation, performed as described above, after removing the planktonic cells. After another 24 h of incubation, the planktonic cells were removed and 150 μL of PBS and 30 μL of 0.3% MTT were added. The microplates were then incubated for 2 h at 37 °C (35 °C for *A. baumannii*). The MTT solution was removed, followed by two washing steps with 200 μL of sterile physiological solution. Finally, 200 μL of dimethyl sulfoxide (DMSO) was added to suspend the formazan crystals and the absorbance was measured at λ = 570 nm (Cary Varian, Palo Alto, CA, USA).

### 4.4. Anti-Enzymatic Activity

#### 4.4.1. Cholinesterases Inhibition

The cholinesterase inhibition was evaluated using Ellman’s colorimetric method [47] with some modifications. Briefly, in a total volume of 1 mL, 415 µL of Tris-HCl buffer 0.1 M (pH 8), 10 µL of a buffer solution of the EOs (in methanol) at different concentrations (100, 10, 1, and 0.1 mg/mL), and 25 µL of a solution containing 0.28 U/mL of AChE (or BChE) were incubated for 15 min at 37 °C. Then, a solution of AChI (or BChI) 1.83 mM (75 µL) and 475 µL of DTNB was added, and the final mixture was incubated for 30 min at 37 °C. The absorbance was measured at 405 nm in a spectrophotometer (Thermo Scientific Multiskan GO, Monza, Italy). Galantamine was the positive control. All experiments were carried out in triplicate and the results are expressed as the mean ± SD.

#### 4.4.2. α-Amylase Inhibition Assay

Amylase activity was determined using the method of Bernfeld [48] with slight modification. An amount of 100 hundred μL of different concentrations of the EOs was added to 200 µL of 20 mM sodium phosphate buffer (pH = 6.9) and 100 µL of amylase solution (10 U/mL). The mixture was incubated at 37 °C for 10 min. Then, 180 µL of 1% soluble starch solution was added and incubated at 37 °C for 20 min. An amount of 180 µL of 3,5 dinitrosalycyclic acid (DNSA) water solution (96 mM) was added to the mixture and boiled in a block heater at 100 °C for 10 min. Then, the solution was cooled by adding 600 µL of distilled water. The absorbance of the solution was read at 540 nm in a UV Spectrophotometer (Thermo Fischer Scientific, Vantaa, Finland). All experiments were carried out in triplicate and the results are expressed as the mean ± SD.

#### 4.4.3. α-Glucosidase Inhibition Assay

α-Glucosidase inhibitory activity was evaluated as previously reported [49] with some modifications. Briefly, the assay was carried out in 96-multiwell plates and 0.1 M of phosphate buffer at pH 7.0 (150 μL) was added to each well; successively, 10 μL of the EOs dissolved in methanol to obtain different concentrations was added to each well. Then, the reaction was initiated by the addition of 15 μL of the α-glucosidase enzyme water solution (1 U/mL) in each well, and the plate was incubated at 37 °C; after 5 min, 75 μL of the substrate (2.0 mM) 4-nitrophenyl α-D-glucopyranoside was added and, successively, the plate was incubated for 10 min at 37 °C. The absorbance was measured at 405 nm in a UV Spectrophotometer (Thermo Fischer Scientific, Vantaa, Finland). The positive control was acarbose. The negative control absorbance (phosphate buffer in place of the sample) was also recorded. Inhibition of the enzyme was calculated and the results are expressed as IC_50_.

The percent of inhibition of the enzyme activity for cholinesterases, α-amylase, and α-glucosidase was calculated by comparison with the absorbance of the control without sample, following the formula:% = [(A0 − A1)/A0] * 100
where A0 is the absorbance of the control without the sample and A1 is the absorbance of the sample. The IC_50_ value was obtained by plotting the inhibition percentage against sample concentrations.

All experiments were carried out in triplicate and the results are expressed as the mean ± SD.

### 4.5. Statistical Analysis

The experiments were replicated three times and the resulting data were subjected to statistical analysis using SPSS statistical software version 26.

## 5. Conclusions

This study contributes to the existing knowledge regarding the composition and bioactivity of various *Eucalyptus* EOs. The EO compositions, coupled with their enzymatic potential and antimicrobial activities, offer insight into the multifaceted attributes of these natural compounds. In particular, the *E. extensa* EO could be taken into consideration for the development of possible antimicrobial agents against *E. coli* and *L. monocytogenes* human infections. Moreover, further investigations can shed more light on the diverse applications of *Eucalyptus* EOs, enhancing our understanding of their potential benefits and expanding their potential applications in various sectors. In fact, beyond their potential use as antimicrobial agents, *Eucalyptus* EOs (especially *E. brevifolia* and *E. leptophylla* EOs) could be used as coadjutants in preventing and treating Alzheimer’s disease and/or diabetes mellitus.

## Figures and Tables

**Figure 1 molecules-28-07211-f001:**
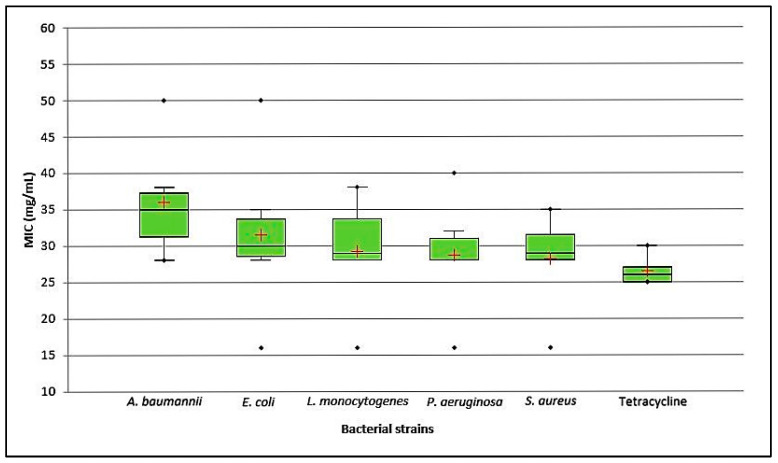
Box–whisker graph representing MIC of six Eucalyptus EOs.

**Figure 2 molecules-28-07211-f002:**
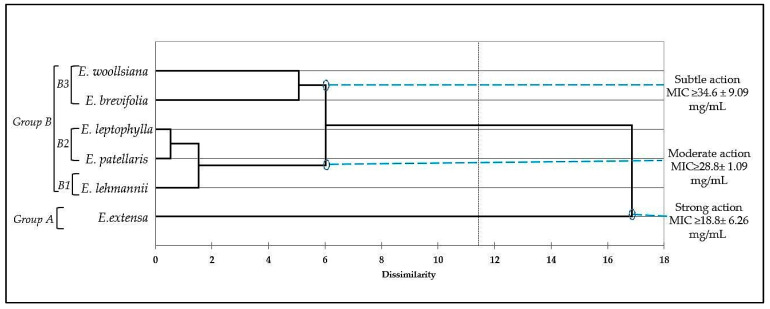
Cluster analysis dendrogram of essential oil antibacterial activity based on Euclidean distance.

**Table 1 molecules-28-07211-t001:** Chemical composition (%, *v*/*v*) of the EOs.

	A	B	C	D	E	F	KI ^a^	KI ^b^	Identification ^c^
3-Ethyl-2-pentanone	-	-	-	0.2	-	-	783	-	1, 2
3-Methylbutyl acetate	-	-	-	0.2	-	-	815	1132	1, 2
4-Methyl-2-pentyl acetate	-	-	-	-	-	0.2	831	-	1, 2
2-Methylbutanoic anhydride	-	-	-	31.0	-	-	833	-	1, 2
α-Pinene	16.6	12.0	21.4	0.7	1.3	7.5	857	1012	1, 2, 3
2,4-Dimethyl-3-ethylpentane	-	-	-	0.3	-	-	859	-	1, 2
Camphene	1.0	0.2	0.9	0.1	-	0.3	868	1075	1, 2, 3
β-Pinene	0.2	0.3	-	-	0.3	0.7	891	1110	1, 2, 3
Myrcene	-	0.2	-	-	0.6	0.1	893	1166	1, 2
α-Phellandrene	0.1	1.0	-	-	1.5	0.1	918	1177	1, 2, 3
2-Ethyl-1-pentanol	-	-	-	0.1	-	-	921	-	1, 2
α-Terpinene	-	0.1	0.2	-	0.4	-	930	1170	1, 2, 3
*p*-Cymene	-	-	-	-	1.8	-	938	1250	1, 2
Eucalyptol	57.0	50.0	70.5	32.2	26.3	53.6	943	1210	1, 2, 3
*cis*-β-Ocimene	0.3	0.1	-	-	-	0.1	955	1225	1, 2
γ-Terpinene	1.1	0.1	-	-	1.0	0.3	971	1221	1, 2, 3
*trans*-*p*-Mentha-2,8-dienol	0.1	-	-	-	-	-	981	-	
Fenchone	0.2	-	-	-	-	-	994	-	
Terpinolene	-	0.2	-	-	0.5	-	996	1267	1, 2, 3
6-Camphenol	0.3	-	-	0.1	-	0.2	998	-	1, 2
*p*-Cymenene	-	0.1	-	-	-	-	999	1269	1, 2
3-Methylbutyl 2-methylbutanoate	-	-	-	0.1	-	-	1008	-	1, 2
Linalool	-	-	-	-	0.4	-	1009	1506	1, 2, 3
α-Pinene oxide	0.1	-	-	-	0.1	0.1	1012	1384	1, 2
Butanoic acid, 3-methyl-, 3-methylbutyl ester	0.2	-	-	0.5	0.2	0.7	1015	1285	1, 2
*endo*-Fenchol	-	-	-	-	0.1	0.6	1016	-	1, 2, 3
*exo*-Fenchol	1.2	0.3	1.3	0.2	-	-	1019	-	1, 2, 3
*cis*-*p*-Menth-2-en-1-ol	-	-	-	-	0.3	-	1024	-	1, 2
α-Campholenal	0.2	0.1	-	-	-	0.1	1027	-	1, 2
2,4,4-Trimethylcyclopentanol	-	-	-	0.1	-	-	1029	-	1, 2
*allo*-Ocimene	0.3	0.1	-	-	-	0.1	1036	1388	1, 2
*trans*-Pinocarveol	5.8	2.1	-	4.6	0.1	4.6	1040	1664	1, 2
*trans*-*p*-Menth-2-en-1-ol	-	-	-	-	0.3	0.1	1042	1571	1, 2
Camphor	0.2						1043	1491	1, 2, 3
Camphene hydrate	-	0.2	-	-	-	-	1046	-	1, 2
*trans*-Pinocamphone	0.2						1049	-	1, 2
Pinocarvone	1.7	0.6	-	0.9	-	-	1065	-	1, 2
Borneol	1.9	0.3	0.9	0.3	0.3	-	1067	1715	1, 2, 3
*neo-iso*-Isopulegol	0.1	0.2	-	-	-	-	1068	-	1, 2
Pinocampheol	0.3						1069	-	1, 2
Terpinen-4-ol	0.6	0.5	-	-	2.1	-	1078	1636	1, 2, 3
Cryptone	-	-	-	-	0.4	-	1084	1659	1, 2
*trans*-p-Mentha-1(7),8-dien-2-ol	-	0.4	-	0.2	-	-	1088	1810	1, 2
α-Terpineol	4.1	1.6	0.1	-	1.6	0.6	1090	1662	1, 2, 3
*cis*-Piperitenol	-	-	-	-	0.2	-	1091	-	1, 2
Dihydrocarveol	-	-	-	-	-	0.4	1092	-	1, 2, 3
Myrtenol	0.3	-	-	0.2	-	0.7	1093	-	1, 2
Safranal	-	-	-	-	0.1	-	1095	1648	1, 2
Verbenone	-	-	-	-	0.2	-	1097	-	1, 2
*trans*-Piperitenol	-	-	-	-	0.3	-	1102	-	1, 2
Pulegone	-	-	-	-	0.2	-	1111	1662	1, 2
*endo*-Fenchyl acetate	0.1	-	-	-	-	-	1112	-	1, 2
*trans*-Chrysantenyl acetate	-	-	-	-	0.2	-	1114	-	1, 2
Isobornyl acetate	0.1	-	-	-	-	-	1118	1582	1, 2
*p*-Menth-8-en-2-ol	0.6	-	-	-	-	-	1226	-	1, 2
Carveol	-	-	-	0.1	-	0.1	1131	-	1, 2, 3
Cumin aldehyde	-	-	-	-	0.2	-	1138	1802	1, 2
*cis*-Ocimenone	-	-	-	-	-	0.6	1139	1225	1, 2
Piperitone	-	-	-	-	0.1	-	1144	1748	1, 2
*trans*-Piperitenone oxide	-	-	-	0.4	-	-	1148	-	1, 2
Phellandral	-	-	-	-	0.3	-	1164	1720	1, 2
Bornyl acetate	0.2						1176	1575	1, 2
Carvacrol	-	-	-	-	0.7	-	1191	2219	1, 2, 3
*p*-Cymen-7-ol	-	-	-	-	-	0.5	1197	2113	1, 2
δ-Elemene	-	0.3	-	-	0.2	0.2	1208	1479	1, 2
Piperitenone	-	0.1	-	-	0.1	-	1215	-	1, 2
Elixene	-	-	-	-	0.9	-	1218	-	1, 2
*exo*-2-Hydroxycineole acetate	0.3	-	-	-	0.1	-	1226	-	1, 2
α-Terpinyl acetate	-	2.4	-	-	-	-	1232	1685	1, 2
Isoledene	-	-	-	-	0.1	-	1251	-	1, 2
α-Copaene	-	0.3	-	-	-	-	1253	1477	1, 2
β-Elemene	-	-	-	-	0.3	-	1275	-	1, 2
Longifolene	-	-	-	0.1	0.3	-	1283	1575	1, 2
*trans*-Caryophyllene	-	0.1	-	-	0.1	0.2	1284	1617	1, 2
α-Gurjunene	-	0.2	-	-	-	0.1	1285	1535	1, 2
β-Gurjunene	-	-	-	-	-	0.2	1298	1655	1, 2
β-Cedrene	-	-	-	-	0.9	-	1299	1613	1, 2
β-Copaene	-	-	-	0.1	0.1	-	1301	-	1, 2
α-Panasinsene	0.1	-	-	-	-	-	1307	-	1, 2
α-Guaiene	-	-	-	-	1.5	-	1307	1600	1, 2
Aromadendrene	-	2.9	-	4.4	-	4.7	1308	1631	1, 2
*allo*-Aromadendrene	-	0.9	-	0.5	-	1.1	1312	1660	1, 2
*cis*-Cadina-1(6).4-diene	-	-	-	-	0.1	-	1322	-	1, 2
Dauca-5,8-diene	-	-	-	-	0.1	-	1325	-	1, 2
*trans*-Cadina-1(6),4-diene	-	-	-	-	0.2	-	1332	-	1, 2
γ-Gurjunene	-	0.6	-	0.1	-	0.6	1335	-	1, 2
β-Selinene	-	-	-	0.2	-	-	1343	1725	1, 2
Guaia-1(10),11-diene	-	0.1	-	-	-	-	1344	-	1, 2
Viridiflorene	-	0.8	-	0.2	-	-	1350	-	1, 2
α-Selinene	-	-	-	0.1	-	-	1351	1713	1, 2
9-*epi*-β-Caryophyllene	-	-	-	-	-	0.4	1352	-	1, 2
Eudesma-4(14),11-diene	-	-	-	-	0.1	-	1353	1708	1, 2
6-[1-(Hydroxymethyl)vinyl]-4,8a-dimethyl-1,2,3,5,6,7,8,8a-octahydro-2-naphthalenol	-	-	-	-	0.7	-	1356	-	1, 2
Longifolene	0.1	-	-	-	-	-	1364	1574	1, 2
*dehydro*-Aromadendrene	-	0.4	-	-	0.4	-	1366	-	1, 2
γ-Muurolene	-	0.1	-	-	-	0.1	1376	1725	1, 2
*trans*-Muurola-4(14),5-diene	-	-	-	-	-	0.4	1381	-	1, 2
*trans*-Cycloisolongifol-5-ol	-	-	-	-	0.1	-	1387	-	1, 2
Cadina-3,9-diene	-	0.2	-	-	-	-	1392	-	1, 2
γ-Vetivenene	-	-	-	-	0.2	-	1411	-	1, 2
α-Amorphene	-	-	-	-	-	1.3	1419	1750	1, 2
Macrocarpal	-	-	-	-	0.5	-	1420	-	1, 2
10-*epi*-Cubebol	-	-	-	3.4	-	4.6	1421	-	1, 2
*trans*-β-Guaiene	-	0.5	-	-	-	-	1425	1651	1, 2
δ-Selinene	-	-	-	-	-	0.6	1426	-	1, 2
Isoaromadendrene epoxide	-	-	-	-	0.3	-	1431	-	1, 2
γ-Patchoulene	-	-	-	-	-	1.3	1436	-	1, 2
Germacrene B	-	-	-	10.2	-	-	1437	1795	1, 2
Spathulenol	0.1	2.3	-	-	28.3	-	1438	2127	1, 2
Globulol	-	3.9	-	1.2	-	0.1	1442	2104	1, 2
Epiglobulol	0.1	-	-	-	0.6	-	1444	-	1, 2
α-Cadinene	-	-	-	-	-	1.3	1450	-	1, 2
Cubeban-11-ol	-	0.3	-	0.6	-	-	1453	-	1, 2
Guaiol	0.1	-	-	-	-	-	1454	2094	1, 2
Rosifoliol	-	0.5	-	1.0	-	0.5	1461	-	1, 2
*cis*-Cadin-4-en-7-ol	-	-	-	-	-	0.3	1473	-	1, 2
Khusimone	-	0.6	-	-	-	-	1474	-	1, 2
9,11-epoxy-Guaia-3,10(14)-diene	-	-	-	0.6	-	-	1475	-	1, 2
γ-Eudesmol	-	2.9	-	0.1	-	-	1477	2178	1, 2
1,7-*diepi*-α-Cedrenal	-	-	-	0.1	-	-	1493	-	1, 2
Cubenol	-	-	-	-	0.3	0.4	1484	2080	1, 2
β-Eudesmol	-	3.2	-	0.8	3.9	1.6	1485	2248	1, 2
Cedr-8(15)-en-9-α-ol	-	-	-	-	2.0	-	1487	-	1, 2
α-Eudesmol	-	2.8	-	0.6	-	0.4	1489	2247	1, 2
α-Cadinol	-	-	-	-	0.7	-	1490	2224	1, 2
Selin-11-en-4-α-ol	-	-	-	-	0.3	-	1492	2273	1, 2
Murolan-3,9(11)-diene-10-peroxy	-	-	-	0.2	-	-	1497	-	1, 2
7-*epi*-α-Eudesmol	-	-	-	-	1.8	-	1506	-	1, 2
Germacra-4(15),5,10(14)-trien-1-α-ol	-	-	-	-	0.8	-	1521	-	1, 2
Corymbolone	-	-	-	0.1	-	-	1567	-	1, 2
Eudesma-4(15),7-dien-1β-ol	-	-	-	-	0.2	-	1571	-	1, 2
Cyperotundone	-	-	-	-	3.0	-	1584	-	1, 2
Eudesm-7(11)-en-4-ol	-	-	-	-	1.9	-	1586	-	1, 2
1,1,4,6-Tetramethyldecahydro-1H-cyclopropa[e]azulene-4,5,6-triol	-	-	-	0.2	-	-	1587	-	1, 2
*epi*-Cyclolorenone	-	-	-	-	0.8	-	1602	-	1, 2
Stigmasterol acetate	0.1	-	-	-	-	-	2735	-	1, 2
Stigmastan-3,5,22-trien	0.1	-	-	-	-	-	2748	-	1, 2
Stigmast-5-en-3-ol, oleate	0.6	-	-	0.2	0.1	-	2769	-	1, 2
Total	97.2	97.4	95.3	97.5	94.1	97			
Monoterpene hydrocarbons	19.6	14.1	22.5	0.8	7.4	9			
Oxygenated monoterpenes	75.7	58.8	72.8	39.2	34.7	67			
Sesquiterpene hydrocarbons	0.2	8.0	-	15.9	5.5	12.2			
Oxygenated sesquiterpenes	0.7	16.5	-	8.9	46.2	7.9			
Others	1.0	-	-	32.7	0.3	0.9			

A = *E. brevifolia*, B = *E. extensa*, C = *E. lehmanii*, D = *E. leptophylla*, E = *E. patellaris*, and F = *E. woolsiana*. ^a,b^ The Kovats retention indices are relative to a series of n-alkanes (C_10_–C_35_) in the apolar HP-5 MS and the polar HP Innowax capillary columns, respectively. ^c^ Identification method: 1 = comparison of the Kovats retention indices with published data; 2 = comparison of mass spectra with those listed in the NIST 02 and Wiley 275 libraries and with published data; and 3 = coinjection with authentic compounds. t = trace (<0.1%). - = absent.

**Table 2 molecules-28-07211-t002:** Inhibitory activity of the EOs on mature biofilm.

	Doses (mg/mL)	* A. baumannii *	* E. coli *	* L. monocytogenes *	* P. aeruginosa *	* S. aureus *
* E. brevifolia *	10	0.00 ± 0.00	70.91 ± 3.47 ^d^	73.53 ± 5.07 ^d^	39.71 ± 2.78 ^d^	0.00± 0.00
	20	0.00 ± 0.00	77.30 ± 1.67 ^d^	87.38 ± 1.23 ^d^	44.25 ± 2.67 ^d^	35.69 ± 3.08 ^d^
* E. extensa *	5	0.00 ± 0.00	30.89 ± 2.67 ^d^	55.10 ± 1.25 ^d^	57.11 ± 4.45 ^d^	9.94 ± 0.78 ^d^
	10	0.00 ± 0.00	89.76 ± 2.06 ^d^	83.27± 1.36 ^d^	63.61 ± 1.67 ^d^	73.73 ± 3.37 ^d^
* E. lehmannii *	10	0.00 ± 0.00	21.52 ± 1.44 ^d^	14.98 ± 0.76 ^d^	75.38 ± 3.98 ^d^	16.06 ± 1.02 ^d^
	20	34.28 ± 2.24 ^d^	39.08 ± 2.44 ^d^	32.27 ± 2.01 ^d^	78.13 ± 3.35 ^d^	61.15 ± 3.65 ^d^
* E. leptophylla *	10	84.96 ± 1.02 ^d^	54.92 ± 3.52 ^d^	15.40 ± 2.01 ^d^	83.12 ± 2.09 ^d^	58.05 ± 1.76 ^d^
	20	88.91 ± 1.32 ^d^	60.86 ± 5.05 ^d^	60.47 ± 4.54 ^d^	93.17 ± 1.02 ^d^	79.12 ± 2.67 ^d^
* E. patellaris *	10	32.37 ± 2.45 ^d^	56.04 ± 4.12 ^d^	77.34 ± 2.76 ^d^	74.41 ± 2.87 ^d^	72.48 ± 0.01 ^d^
	20	37.63 ± 1.98 ^d^	59.83 ± 2.86 ^d^	88.56 ± 1.08 ^d^	84.70 ± 2.05 ^d^	89.39 ± 3.02 ^d^
* E. woollsiana *	10	0.00 ± 0.00	0.00 ± 0.00	0.00 ± 0.00	12.91 ± 1.67 ^d^	45.89 ± 4.01 ^d^
	20	37.59 ± 2.47 ^d^	6.05 ± 0.33 ^d^	0.00 ± 0.00	15.60 ± 1.54 ^d^	67.27 ± 5.97 ^d^

The results are the mean of three independent experiments ± SD. d: *p* < 0.0001 compared with the positive control (inhibition = 0) according to two-way ANOVA followed by Dunnet’s multiple comparison test at the significance level of *p* < 0.05.

**Table 3 molecules-28-07211-t003:** Inhibitory activity of the EOs on the metabolism of the bacterial sessile cell in mature biofilm.

	Doses (mg/mL)	*A. baumannii*	*E. coli*	*L. monocytogenes*	*P. aeruginosa*	*S. aureus*
*E. brevifolia*	10	18.99 ± 2.76 ^d^	27.46 ± 1.66 ^d^	25.53 ± 2.07 ^d^	20.27 ± 1.89 ^d^	12.21 ± 0.98 ^d^
	20	28.83 ± 1.87 ^d^	78.8 ± 4.07 ^d^	32.46 ± 3.33 ^d^	30.1 ± 2.76 ^d^	76.26 ± 2.65 ^d^
*E. extensa*	5	49.95 ± 4.44 ^d^	17.72 ± 1.01 ^d^	83.86 ± 4.91 ^d^	0.00 ± 0.00	0.00 ± 0.00
	10	65.40 ± 4.57 ^d^	56.63 ± 0.97 ^d^	89.26 ± 3.54 ^d^	45.37 ± 1.33 ^d^	11.30 ± 1.07 ^d^
*E. lehmannii*	10	51.39 ± 4.55 ^d^	33.34 ± 2.91 ^d^	63.36 ± 3.91 ^d^	84.05 ± 1.67 ^d^	59.57 ± 3.67 ^d^
	20	54.35 ± 3.85 ^d^	21.74 ± 1.76 ^d^	64.73 ± 3.91 ^d^	67.77 ± 3.92 ^d^	68.66 ± 2.64 ^d^
*E. leptophylla*	10	54.92 ± 3.14 ^d^	15.41 ± 1.05 ^d^	83.12 ± 1.18 ^d^	58.05 ± 1.66 ^d^	51.23 ± 3.01 ^d^
	20	60.86 ± 3.94 ^d^	60.47 ± 3.08 ^d^	93.17 ± 1.13 ^d^	79.12 ± 2.97 ^d^	81.04 ± 2.76 ^d^
*E. patellaris*	10	26.47 ± 2.01 ^d^	19.48 ± 0.78 ^d^	0.00 ± 0.00	16.44 ± 1.02 ^d^	29.89 ± 1.79 ^d^
	20	58.75 ± 3.02 ^d^	36.12 ± 2.41 ^d^	22.11 ± 2.55 ^d^	37.25 ± 3.07 ^d^	32.21 ± 1.98 ^d^
*E. woolsiana*	10	75.08 ± 4.22 ^d^	32.55 ± 2.26 ^d^	0.00 ± 0.00	0.00 ± 0.00	48.04 ± 5.01 ^d^
	20	93.27 ± 1.01 ^d^	39.58 ± 4.12 ^d^	7.02 ± 1.22 ^d^	10.76 ± 1.01 ^d^	55.59 ± 3.72 ^d^

The results are the average of three independent experiments ± SD. d: *p* < 0.0001 compared with the positive control (inhibition = 0) according to two-way ANOVA followed by Dunnet’s multiple comparison test at the significance level of *p* < 0.05.

**Table 4 molecules-28-07211-t004:** Inhibitory effects of the EOs on AChE, BChE, α-amylase, and α-glucosidase.

EOs	IC_50_ (mg/mL)			
	AChE	BChE	α-Amylase	α-Glucosidase
*E. brevifolia*	0.3 ± 0.04 ^a^	11.86 ± 3.54 ^c^	0.88 ± 0.14 ^a^	27.31 ± 2.11 ^a^
*E. extensa*	0.25 ± 0.01 ^a^	7.37 ± 0.06 ^b^	n.a	n.a
*E. lehmanii*	1.2 ± 0.40 ^b^	3.48± 0.37 ^a^	16.94 ± 2.14 ^b^	n.a
*E. leptophylla*	0.57 ± 0.03 ^a^	4.54 ± 0.11 ^ab^	0.88 ± 0.05 ^a^	29.0 ± 1.32 ^a^
*E. patellaris*	0.60 ± 0.04 ^a^	3.50 ± 0.10 ^a^	0.91 ± 0.01 ^a^	n.a
*E. woollsiana*	0.38 ± 0.01 ^a^	4.49 ± 0.01 ^ab^	11.9 ± 1.3 ^b^	n.a
Galantamine	0.008 ± 0.003	0.05 ± 0.01	-	-
Acarbose	-	-	0.004 ± 0.002	0.6 ± 0.3

AChE: Acetylcholinesterase; BChE: butyrylcholinesterase. The results are the mean ± SD of three experiments. Different letters indicate mean values significantly different at *p* < 0.05 according to a one-way ANOVA followed by Tukey’s post hoc test. n.a = not active (IC_50_ > 30 mg/mL).

**Table 5 molecules-28-07211-t005:** Species, origin, bioclimatic stage, and yields of studied species.

Species	Arboretum (Governorate)	Harvest Period	Bioclimatic Condition	Yields %
*E. brevifolia*	Hajeb Layoun (Kairouan)	April 2023	Semi-arid upper with moderate winters	2.42
*E. extensa*	Souiniet (Ain Draham)	June 2021	Upper humid	1.43
*E. lehmannii*	Korbous (Nabeul)	May 2022	Sub-humid	1.91
*E. leptophylla*	Hajeb Layoun (Kairouan)	April 2023	Semi-arid upper with moderate winters	2.03
*E. patellaris*	Hajeb Layoun (Kairouan)	April 2023	Semi-arid upper with moderate winters	0.25
*E. woolsiana*	Djebel Mansour (Zaghouan)	Mars 2021	Semi-arid	1.66

## Data Availability

The data presented in this study are available on request from the corresponding author.

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
