# Peer review of "Chemical Composition, Antibacterial Properties, and Anti-Enzymatic Effects of Eucalyptus Essential Oils Sourced from Tunisia"

_molecules, 2023, doi:10.3390/molecules28207211_

Round 1
Reviewer 1 Report
The manuscript title: “Chemical Composition, Antibacterial Properties, and Anti-Enzymatic Effects of Eucalyptus Essential Oils sourced from Tunisia”
This study examined the chemical composition of six Eucalyptus essential oils (EOs) from different species and evaluated their anti-enzymatic and antibiofilm activities.
Comments and Suggestions for Authors
-Following the study's title, the results and discussion should be presented in the following order: chemical composition, antibacterial properties, and anti-enzymatic Effects.
- The authors' study on six essential oils should maintain a consistent pattern when presenting the results, starting with the same format of Eucalyptus species for ease of understanding, for example E. brevifolia, E. extensa, E. lehmanii, E. leptophylla, E. patellaris, E. woollsiana
Introduction
Introduction should be improved
Line 67-69: With its extensive diversity of species and their diverse biological activities, like antioxidant
[6], anti-diabetic [7], phytotoxic [8], antimicrobial [9] and insecticidal [10]. Please put more detail in antimicrobial.
Results
-The results should present follow the title, start with chemical compositions, antibacterial properties, and anti-enzymatic Effects.
-Table 2: AChE, BChE should write the full name to explain under the table
-Line 185-188: the authors mention that EOs rich in oxygenated sesquiterpenes like globulol, pathulenol, and β-eudesmol, exemplified by E. extensa (16.5%) and E. patellaris (46.2%), exhibited the lowest MIC values. Therefore, the discussion section should include a discussion of the activity of oxygenated sesquiterpenes
Discussion
- While the authors provided a detailed discussion of chemical compositions, it can be improvement in connecting these chemical compositions to bioactivities such as antibacterial properties and anti-enzymatic effects.
- The authors should discuss essential oils that could be candidates for antibacterial and anti-enzymatic effects.
- E. extensa essential oil that has the best antimicrobial activity and their chemical composition should be discussion.
Others:
- Please check for consistency in the use of 'antibiofilm,' 'hr,' 'µL', and 'min' throughout the manuscript.
-Line 38: what is “AD” mean
-Line 45: in this context, à In this context,
-Line 344-347: Is “1,8-cineole” the same as “eucalyptol”
-Line 496: revise (5 × 106 cfu/mL) to (5 × 106 CFU/mL)
-Line 448: “One hundred µiL” à One hundred µL

Reviewer 2 Report

A review of the English language by a native speaker is advisable
Author Response
Dear Editor,
please, see the attachment
